# Achieving Tunable Microwave Absorbing Properties by Phase Control of NiCoMnSn Alloy Flakes

Xiaogang Sun [1,2], Jian Xu [1], Lian Huang [1,2,*], Daitao Kuang [3,*], Jinrong Liu [1], Guanxi Wang [1], Qifei Zhang [1] and Yonghua Duan [4]

1   Hunan Institute of Engineering, College of Mechanical Engineering, Xiangtan 411104, China
2   Hunan Engineering Research Center of New Energy Vehicle Lightweight, Xiangtan 411104, China
3   School of Computational Science and Electronics, Hunan Institute of Engineering, Xiangtan 411104, China
4   Faculty of Material Science and Engineering, Kunming University of Science and Technology, Kunming 650093, China
*   Correspondence: huanglianfriend@163.com (L.H.); kdt_hnie@163.com (D.K.)

**Abstract:** Microwave absorption performance of metal alloys are highly dependent on their phase structures. However, the phase control of Ni–Mn-based alloys to achieve effective microwave absorption properties has been rarely reported. In this work, $Ni_{43}Co_7Mn_{39}Sn_{11}$ alloy flakes were fabricated by balling milling method, and the contents of $\gamma$ phase in the flakes were tuned by the subsequent heat treatment process. The as-fabricated $Ni_{43}Co_7Mn_{39}Sn_{11}$ alloy flakes exhibited excellent tunable microwave absorption by control of their phase structures. The optimal reflection loss was lower, up to $-56.4$ dB at 8.8 GHz, and was achieved at a single thickness of 2.0 mm. This can be attributed to the optimal structure of $Ni_{43}Co_7Mn_{39}Sn_{11}$ alloy flakes by phase control, and thus achieving improved attenuation property and impedance matching. This study proved $Ni_{43}Co_7Mn_{39}Sn_{11}$ alloy flakes should be a promising microwave absorption material. It is also demonstrated that phase control is an effected strategy for optimal microwave absorption properties of metal alloys and may have some reference value for related studies.

**Keywords:** NiCoMnSn alloy flakes; microwave absorption; $\gamma$ phase; heat treatment; impedance matching





## 1. Introduction

Microwave Absorbing Materials (MAMs) have been widely studied in recent decades for preventing electromagnetic interference (EMI) for military requirement, ensuring the safety of electron equipment, and mitigating human health hazards arising due to electromagnetic radiations [1–3]. The ideal MAMs possess the characters of being strong, wide, light, and thin. Due to microwave-absorbing properties being co-determined by attenuation ability and impedance matching, numerous efforts have modified micromorphology, components, and microstructures to balance the attenuation ability and impedance matching [2–5].

As a kind of widely used MAM, carbon-based materials were modified to different microstructures, including black carbon [3], graphite [6], hard carbon [7], nano-carbon tube [8,9], etc., to improve their microwave absorption properties. Metal phase-controlled MAMs were also studied on the Ni-based and Co-based materials [10–12]. The 3 nm HCP-Ni NPs/graphene testified better EMW absorbability than that of the 15 nm FCC-Ni NPs on graphene [10]. HCP-Ni phase structure-based Ni-C nanoparticles demonstrated better microwave absorption performance than that of FCC-Ni-C nanoparticles [12]. Moreover, a similar phase-controlled effect has been revealed in the $\alpha$-Co nanospheres/graphene nanosheets (GN) and $\beta$-Co/GN composites [11].

Ferromagnetic Ni–Mn-based alloys have been widely studied as a novel kind of multifunctional material [13,14]. The magnetic shape memory effect [14], magnetoresistance effect [15], and magnetocaloric effect [16] of Ni–Mn-based alloys correspond to their wealthy

phase structure transformation. However, the effect of modifying phase structure on their microwave absorption performance has not been studied.

In this paper, the content of $\gamma$-phase $Ni_{43}Co_7Mn_{39}Sn_{11}$ alloy flakes was controlled by thermal treatment. The $Ni_{43}Co_7Mn_{39}Sn_{11}$ alloy flakes annealed in vacuum atmosphere for 6 h exhibit excellent microwave performance. The optimal reflection loss was lower, up to $-56.4$ dB at 8.8 GHz, and was achieved at a single thickness of 2.0 mm. It is demonstrated that phase control is an effected strategy for the optimal microwave absorption properties of metal alloys.

## 2. Materials and Methods

A polycrystalline $Ni_{43}Co_7Mn_{39}Sn_{11}$ (at. %) ingot was prepared by arc-melting mothed with high-purity raw elements Ni (4N), Co (4N), Mn (3N), and Sn (4N). For homogenization, the ingot was re-melted four times and then annealed at 900 °C for 48 h under vacuum atmosphere, followed by quenching into ice water. The obtained $Ni_{43}Co_7Mn_{39}Sn_{11}$ ingot was crushed to small particles and then balled on a planetary ball mill at 400 rpm (Nju-instrument Co., Ltd., Nanjing, China). The weight ratio between stainless steel balls and $Ni_{43}Co_7Mn_{39}Sn_{11}$ particles was set to 40:1. Then, ethanol (100 mL) was added to control the temperature and avoid oxidation of the particles. After ten hours of balling, the size of the particles was controlled to fewer than 48 μm by the sieving method. The balled $Ni_{43}Co_7Mn_{39}Sn_{11}$ powders were divided into three parts for no heat-treatment (S-0h) and vacuum atmosphere heat treatment at 600 °C for 6 h (S-6h) and 12 h (S-12h).

The crystal structure of $Ni_{43}Co_7Mn_{39}Sn_{11}$ powders were measured by powder X-ray diffraction (XRD, Bruker D8, Billerica, MA, USA) using Cu K$\alpha$ radiation. Micromorphology and chemical composition were carried out on a Scanning electron microscope (SEM, Hitachi SU3500, Tokyo, Japan). The electromagnetic parameters from 2.0 to 18.0 GHz were detected by a vector Network Analyzer (Agilent N5224B, Santa Clara CA, United States). The flakes were uniformly dispersed in paraffin in a mass ratio of 2:1. The flake–paraffin composites were shaped as a toroidal with an inner diameter of 3.04 mm, outer diameter of 7.0 mm, and thickness of 2.0 mm.

## 3. Results and Discussion

Figure 1 shows the typical microstructure image of NiCoMnSn samples S-0h, S-6h, and S-12h. Being ten-hour balled, most parts of the samples present with thin sheet-like structures and non-smooth surfaces. These samples are defined as NiCoMnSn flakes in the following sections. The irregular flakes are fewer than 48 μm in diameter (Figure 1a,c,e) and 0.5 μm in thickness, as shown in Figure 1f. Different from the smooth surface of FeSiAl particles [17,18], the NiCoMnSn flakes exhibit rugged surfaces, as in Figure 1b,c.

The ball milling process brings about many defects, rugged surfaces, and flaky shapes of NiCoMnSn samples. These characteristics will largely enhance the microwave absorption properties of NiCoMnSn samples. First, the defects will act as the polarization centers, enhancing the polarization and thus the dielectric loss of the samples [19,20]. Second, the rugged surface and flaky shape may increase the microwave reflection and scattering. Third, the flaky shape structure can largely enhance their microwave-absorbing ability at high frequencies by reducing the Snoke's limits [18,21]. The reason can be clarified by Formula (1) following the Landau–Lifshitz–Gilbert (LLG) equation [21,22]:

$$(\mu_s - 1)f_r = \frac{2\gamma'M_s}{3\pi}\sqrt{\frac{H_{ha}}{H_{ea}}} \tag{1}$$

where $\mu_s$ is the initial permeability, $f_r$ is resonance frequency, $\gamma'$ is gyromagnetic ratio, $M_s$ is saturation magnetization, $H_{ha}$ is the out-of-plane anisotropy field, and $H_{ea}$ is the in-plane anisotropy field, respectively. For the rugged surface and flaky shape powders, they have high shape anisotropy, which means $H_{ha} >> H_{ea}$. Therefore, the large $H_{ha}/H_{ea}$ leads to large $\mu_s$ and microwave absorption ability [18,21]. The morphology of the samples with the

rugged surface and flaky shape has not been affect by the annealing thermal treatment, as shown in Figure 1a–f.

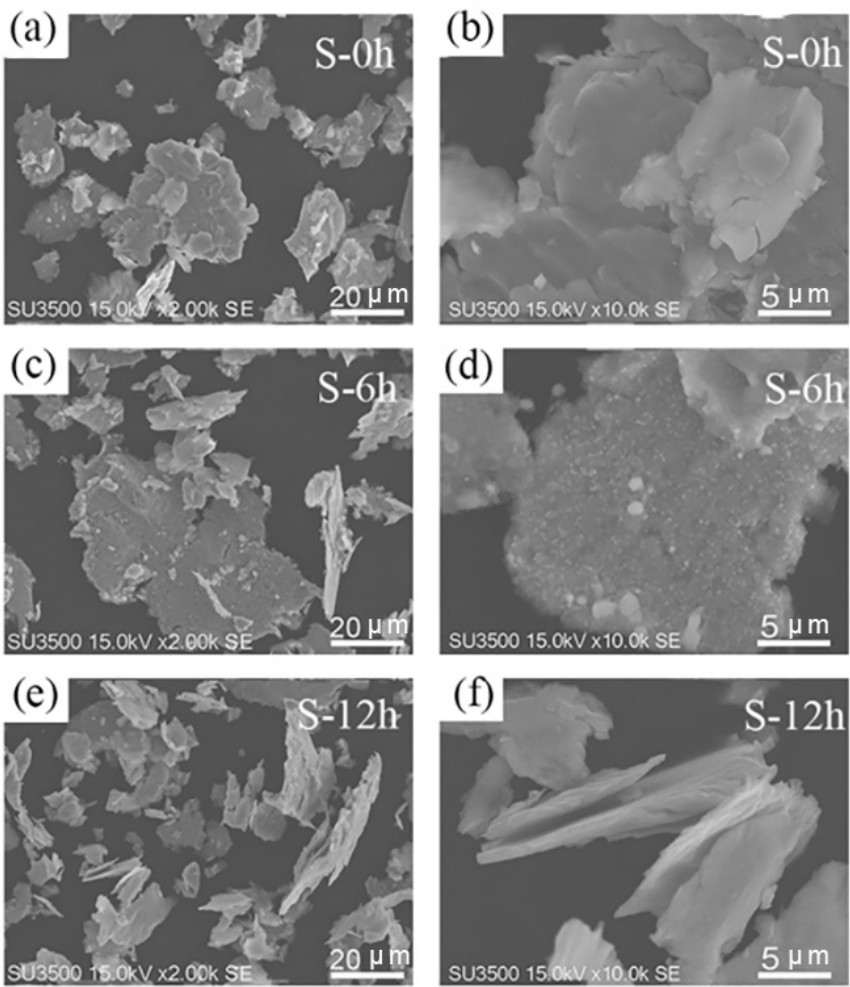

**Figure 1.** SEM image of NiCoMnSn flakes: (**a**,**b**) S-0h, (**c**,**d**) S-6h, and (**e**,**f**) S-12h.

Figure 2 displays the XRD pattern of three samples for NiCoMnSn alloy flakes. The XRD structures of three samples exhibit the mixture of austenite $L2_1$ and $\gamma$ phase. A similar typical eutectic microstructure was also found in $Ni_{38}Co_{12}Mn_{41}Sn_9$ and $Ni_{42.4}Co_{8.7}Mn_{37.8}Sn_{11.1}$ [13,23]. First, the large balling interstress suppresses the reflections of the $L2_1$ phase from the XRD pattern of S-0h. With the balling interstress released by vacuum thermal treatment, S-12h exhibits strong refection of the $L2_1$ phase and weak refection of the $\gamma$ phase. However, the relative intensity of the $\gamma$ phase of S-6h is significantly higher than that of S-12h, and the peak values of the $L2_1$ phase compared are similar to that of S-12h. Unlike the suppression of the $\gamma$ phase in $Ni_{38}Co_{12}Mn_{41}Sn_9$ alloy by melt spinning in other reports [23], the $\gamma$ phase of NiCoMnSn alloy in this work is controlled by the thermal treatment.

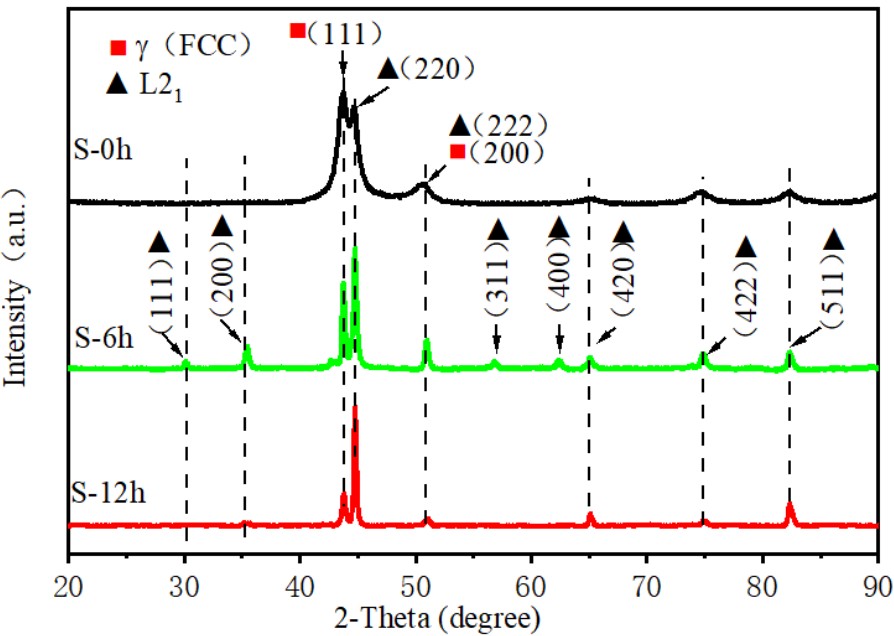

**Figure 2.** XRD pattern of three samples for NiCoMnSn alloy flakes.

Figure 3a,b displays the real permittivity ($\varepsilon'$) and imagined permittivity ($\varepsilon''$) for three samples from 2.0–18.0 GHz, respectively. For all samples, the values of $\varepsilon'$ fluctuate throughout the testing period (i.e., 23–25,18–19 and 14.5–15.5 for S-0h, S-6h, S-12h, respectively). The value fluctuations of $\varepsilon''$ for S-12h and S-6h remain low-level, with 1–2 for the former and 0–0.5 for the latter. The S-0h shows a higher value of $\varepsilon'$ and $\varepsilon''$, but the S-6h exhibits a lower value of $\varepsilon'$ and $\varepsilon''$. However, there is a resonance peak value of about 5.7 at 15.8 GHz for the S-0h, which is similar to the value of $\varepsilon''$ for NiMnGa alloys [4]. The resonance peak can be attributed to the interfacial polarization between the $\gamma$ phase and L2$_1$ phase in S-0h. The typical eutectic microstructure with $\gamma$ phase and L2$_1$ phase would lead to a heterojunction capacitor [23–26]. The heterojunction capacitor from the heterostructures may result in nonlinear resonant behaviors, and similar observation can be found in Fe/C porous nanofibers, C@FeNi3 core/shell, and CdS/$\alpha$-Fe$_2$O$_3$ [24–26]. However, the S-6h and S-12h are absent from the resonance peak. This may be due to the formation of the heterojunction capacitor affected by the content of two different phase structures. The main phase in S-0h is the $\gamma$ phase, while it is the L2$_1$ phase in S-6h and S-12h.

Figure 3c,d shows the real permeability ($\mu'$) and imagined permeability ($\mu''$) for the three samples from 2.0–18.0 GHz, respectively. The values of $\mu'$ for S-12h and S-6h are very close. They rapidly decrease from 1.65 to 0.96 with the frequency increase during the testing period. However, the values of $\mu'$ for S-0h drop from 1.53 to 0.86 with the frequency range of 2.0–15.0 GHz and then shift to 1.04 until 18.0 GHz. The values of $\mu''$ for S-12h gradually decline from 0.5 to 0.25 with the increase in frequency and resemble that of S-6h, except at 2.0–6.0 GHz. On the other hand, the values of $\mu''$ for S-0h rise to 0.38 at 5.0 GHz and then drop to 0 at 15.0 GHz and finally remain around the 0 level to 18.0 GHz.

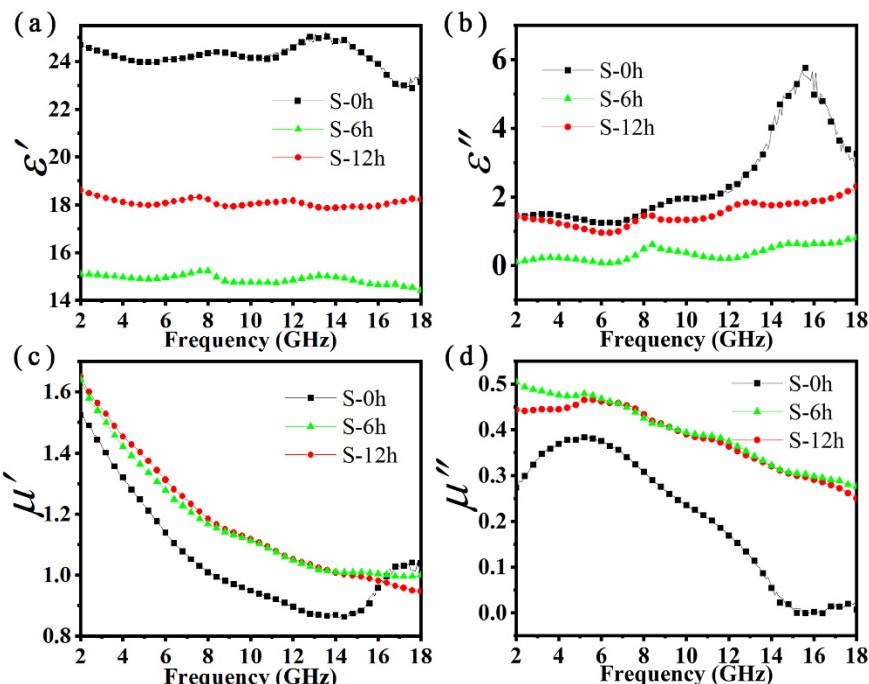

**Figure 3.** Electromagnetic performance for three samples of NiCoMnSn alloy flakes: (**a**) $\varepsilon'$−$f$, (**b**) $\varepsilon''$ −$f$, (**c**) $\mu'$ −$f$, and (**d**) $\mu''$ −$f$.

The values of $\mu'$ for S-0h are less than 1 at high frequency. This phenomenon has been found in many references [27–31]. It can be explained by the formula $\mu' = 1 + (Ms/H) \cos\beta$, where $H$ and $\beta$ represent the intensity of external magnetic field and the phase lag angle of magnetization behind external magnetic field, respectively [1]. The resonance behavior may largely lag the external magnetic field at high frequency. Thus, the value of $\cos\beta$ will be a negative number while the phase lag angle is bigger than 90 degrees. The magnetic $L2_1$ phase is the main phase in the S-6h and S-12h, while the nonmagnetic $\gamma$ phase is the main phase in the S-0h. Therefore, only S-0h exhibits this unusual phenomenon. It can be also found that S-0h has higher levels of $\mu'$ and $\mu''$ and lower levels of $\varepsilon'$ and $\varepsilon''$, while S-6h displays the reverse trend of the electromagnetic parameters. This result may be attributed to the different content of the $\gamma$ phase and $L2_1$ phase in these two samples. The difference in the conductivity and magnetism between the two phases may lead to the difference in the electromagnetic parameters.

To explore the absorbing performance of NiCoMnSn alloy flakes, the reflection loss (*RL*) versus frequency and thickness were calculated by Formulas (2) and (3) following the transmission line theory [1,32].

$$Z_{in} = Z_0 \, (\mu_r/\varepsilon_r)^{1/2} \, tanh \, [j \, (2\pi fd/c)(\mu_r/\varepsilon_r)^{1/2}] \tag{2}$$

$$RL = 20\log|(Z_{in} - Z_0)/(Z_{in} + Z_0)| \tag{3}$$

Here, $Z_{in}$ and $Z_0$ represent the input impedance at the interface of the absorber and the impedance of free space, $\mu_r$ and $\varepsilon_r$ are the electromagnetic parameters of complex permeability and complex permittivity, $f$ is the frequency of microwave, $d$ is the thickness of the absorber, and $c$ is the velocity of light in free space, respectively. Generally, *RL* values of −10 dB correspond to 90% attenuation ability of the incident microwave. Therefore, the effective absorption bandwidth (the frequency range for $RL \leq −10$ dB) is abbreviated as $BW_{eff}$ from a practical point of view [2,12,33].

Figure 4 depicts the three-dimensional (3D) *RL* versus frequency and thickness for the three samples of NiCoMnSn flake–paraffin composites, respectively. As can be seen from Figure 4, NiCoMnSn shows excellent absorbing performance in a broad frequency range of

2.2–14.4 GHz, which includes all of C-band and Ku-band. The corresponding $RL_{min}$ values are −34.7 dB at 3.2 GHz with 4 mm for S-0h, -56.4 dB at 8.8 GHz with 2 mm for S-6h, and 38.2 dB at 4.9 GHz with 3 mm for S-12h, respectively. Moreover, the $BW_{eff}$ for NiCoMnSn flakes is 10.2 GHz (S-0h), 12.9 GHz (S-6h), and 12.8 GHz (S-12h) by calculating the different distances of the dotted line across $RL \leq −10$ dB from Figure 6. Therefore, annealing treatment with 6 h improves the microwave absorption performance, which presents: (i) enhanced amplitude of $RL$ peaks from −34.7 dB to −56.4 dB, (ii) enlarged absorption bandwidth with thinner thickness from 4 mm to 2 mm, and (iii) alerted peak frequency from 3.2 GHz to 8.8 GHz. For comparison, the microwave absorption parameters of NiCoMnSn alloy flakes are listed in Table 1 with other results of ferromagnetic alloy MAMs.

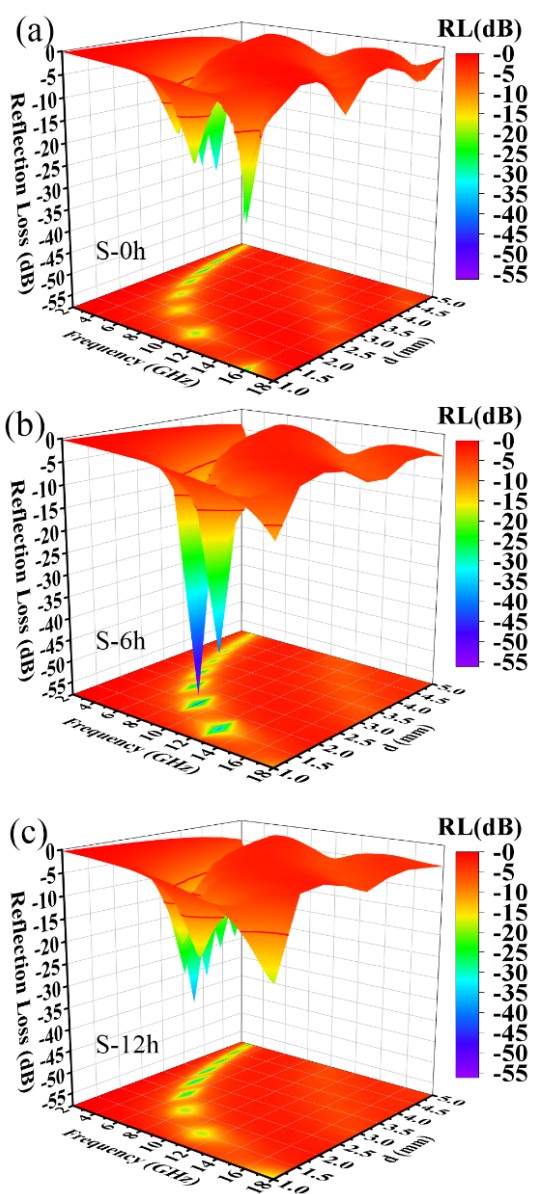

**Figure 4.** $RL-$3D curves of NiCoMnSn alloy flakes samples: (**a**) S-0h, (**b**) S-6h, and (**c**) S-12h.

**Table 1.** A comparison with the microwave absorption parameters of the typical ferromagnetic MAMs with paraffin matrix.

| Absorbers | $RL_{min}$ (dB) | $t_m$ (mm) | $BW_{eff}$ (GHz) | Reference |
|---|---|---|---|---|
| S-0h | −34.7 at 3.2 GHz | 4.0 | 10.2 (2.2–8.2,9.2–11.6,15.3–17.1) | This work |
| S-6h | −56.4 at 8.8 GHz | 2.0 | 12.9 (2.5–14.4,17–18) | This work |
| S-12h | −38.2 at 4.9 GHz | 3.0 | 12.8 (2.3–12.8,15.7–18) | This work |
| $Ni_2MnGa$ | −49.1 at 13.3 GHz | 2.57 | 12.7 (6.3–18) | [5] |
| $Ni_2MnGa$ | −65.2 at 14.4 GHz | 2.9 | 12.7 (6.3–18) | [4] |
| $Fe_{0.5}Co_{0.5}$ | −46.7 at 8.47 GHz | 2.0 | 12.6 (5.4–18.0) | [34] |
| $Fe_{0.6}Co_{0.4}$ | −48.9 at 7.12 GHz | 2.5 | 12.5 (4.2–16.7) | [34] |
| $Fe_{73.5}Si_{13.5}Nb_3Cu_1B_9$ | −45.97 at 10.03 GHz | 2.6 | / | [35] |
| $Fe_{2.25}Co_{72.75}Si_{10}B_{15}$ | −47.9 | / | / | [36] |

Since the microwave absorption performance of MAMs is bound up with the normalized impedance matching $Z_{in}/Z_0$ and the attenuation constant ($\alpha$), the $Z_{in}/Z_0$ and $\alpha$ versus frequency curves of three samples are shown in Figure 5a,b. The values of $Z_{in}/Z_0$ from the S-6h approach to 1 across a wider frequency range. The value of $Z_{in}/Z_0$ close to 1 indicates the ideal impedance matching behavior, which is the efficient complementarity between the relative permittivity and permeability [37]. As shown in Figure 5c, the dielectric loss tangent $tan\delta_\varepsilon$, and in Figure 5d the magnetic loss tangent $tan\delta_\mu$ of three samples, the S-6h exhibits the lowest $tan\delta_e$ and highest $tan\delta_\mu$. For metal electromagnetic absorption materials, balancing between their permittivity and permeability will effectively improve their absorption properties due to better impedance matching [17,18,21]. Thus, modifying the permittivity and permeability of NiCoMnSn alloys flakes will enhance their microwave absorption properties.

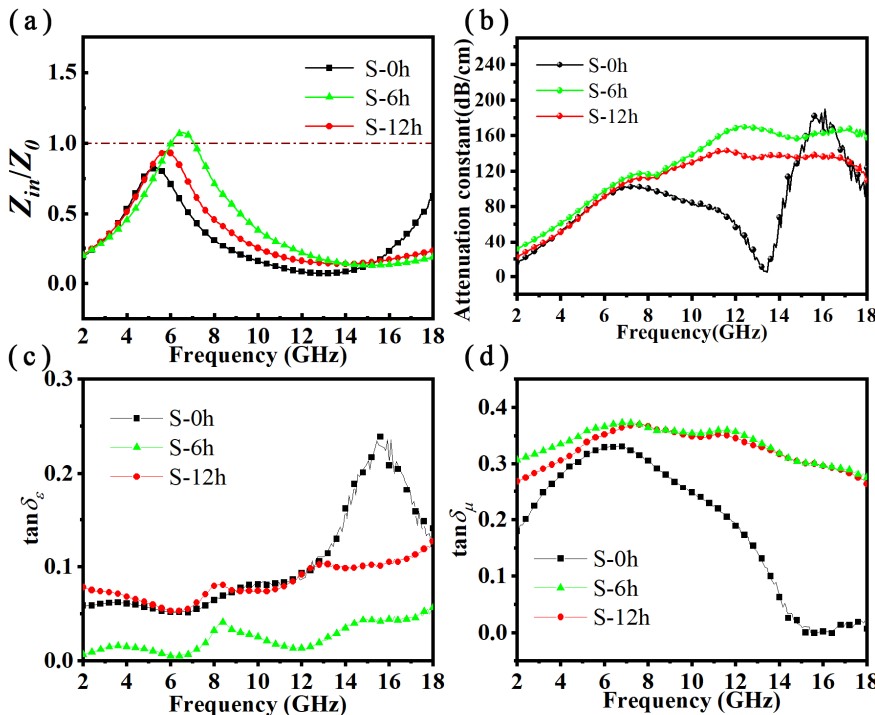

**Figure 5.** (a) $Z_{in}/Z_0 - f$, (b) $\alpha - f$, (c) $tan\delta_\varepsilon - f$, and (d) $tan\delta_\mu - f$ of three samples for NiCoMnSn alloy flakes.

In addition, the attenuation constant ($\alpha$) can be expressed from the electromagnetic parameters by the following Formula (4) [7,32]:

$$\alpha = \sqrt{2}\pi f \times c^{-1} \times \left[ (\mu''\varepsilon'' - \mu'\varepsilon') + \sqrt{(\mu''\varepsilon'' - \mu'\varepsilon')^2 + (\varepsilon'\mu'' + \varepsilon''\mu')^2} \right]^{\frac{1}{2}} \tag{4}$$

First, the $\alpha$ of the three samples is close, from 2 GHz to approximate 7 GHz, with a slow increase to 105 dB/cm due to its $tan\delta_\varepsilon$ and $tan\delta_\mu$ with a small difference. However, the $\alpha$ of S-0h decreases to zero at 13.5 GHz, corresponding to its $tan\delta_\mu$ and drops to zero along the $tan\delta_e$ increases. Finally, the $\alpha$ of S-0h increases to a peak of 185 dB/cm at 15.5 GHz, which interestingly corresponds to the same frequency of the $tan\delta_\varepsilon$ peak, although its $tan\delta_\mu$ fluctuates at the zero level. Second, the $\alpha$ of S-12h and S-6h increases to 8.5 GHz with similar values, then the former rises to 170 dB/cm at 12.5 GHz, and the latter goes to 140 dB/cm at 12 GHz. Therefore, the changes of the $\alpha$ of S-12h and S-6h correspond to their incremental $tan\delta_\varepsilon$ and fluctuant $tan\delta_\mu$, from 2 GHz to 18 GHz. Obviously, the $tan\delta_\mu$ of S-12h and S-6h are larger than their $tan\delta_\varepsilon$, revealing that the main contribution of the absorption ability results from their magnetic loss.

The quarter-wavelength ($\lambda/4$) cancellation model is applied to interpret that the $RL_{min}$ decreases to low frequency as the thickness increases. From this model, the $RL_{min}$ can be obtained at given frequencies if the thickness of absorbers satisfies [7,33]:

$$d = nc / \left[ \frac{4f_m}{(|\mu_r||\varepsilon_r|)^{1/2}} \right] \quad (n = 1, 3, 5, \ldots) \tag{5}$$

Here, $f_m$ represents the peak frequency of $RL$. Figure 6 depicts the thickness values ($d_{cal}$) calculated by Formula 5 with $n = 1$ and the measured peak thickness values ($d_m$) for $RL_{min}$ of NiCoMnSn alloy flakes. Obviously, the given thickness ($d_m$) of the three samples is in good accord with the calculation curve of $d_{cal}$. It shows that the experimental data (black stars) are located around the $\lambda/4$ curves, suggesting that the phase difference between the incident EM waves and the EM waves reflected at the air/media interface is 180°, resulting in the cancellation interference of EM waves to achieve higher $RL$ values [38].

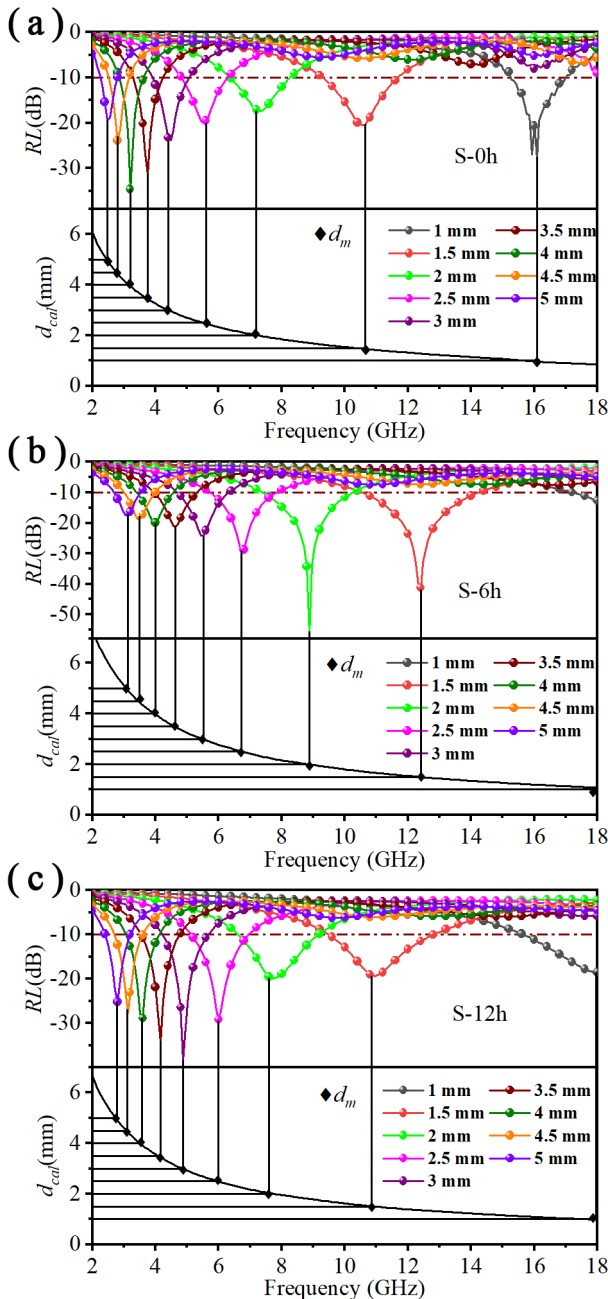

**Figure 6.** *RL−f* curves and the corresponding $d_m−f$ and $d_{cal}−f$ ($n = 1$) curves of NiCoMnSn alloy flakes: (**a**) S-0h, (**b**) S-6h, and (**c**) S-12h.

## 4. Conclusions

Ni$_{43}$Co$_7$Mn$_{39}$Sn$_{11}$ alloy flakes with $\gamma$ and L2$_1$ phases were fabricated by balling milling method. It was found that the metastable $\gamma$ phase would be transformed into L2$_1$ phase after the annealing process. By controlling for the annealing times of 6 h and 12 h, Ni$_{43}$Co$_7$Mn$_{39}$Sn$_{11}$ alloy flakes with decreased $\gamma$ phase were obtained. A systematic study on their MA properties suggested that Ni$_{43}$Co$_7$Mn$_{39}$Sn$_{11}$ alloy flakes exhibited excellent tunable microwave absorption properties by controlling their content of $\gamma$ phase. The optimal reflection loss was lower, up to $-56.4$ dB at 8.8 GHz, and was achieved at a single thickness of 2.0 mm. The as-fabricated Ni$_{43}$Co$_7$Mn$_{39}$Sn$_{11}$ alloy flakes are expected to be an effective microwave absorption material for practical application.

**Author Contributions:** Conceptualization, X.S. and L.H.; methodology, J.X. and G.W.; validation, J.L. and Q.Z.; formal analysis, Y.D. and G.W.; investigation, J.X. and Y.D.; data curation, J.L. and Q.Z.; writing—original draft preparation, X.S.; writing—review and editing, L.H. and D.K.; project administration, X.S. and D.K.; funding acquisition, X.S. and L.H. All authors have read and agreed to the published version of the manuscript.

**Funding:** This research was supported by Natural Science Foundation of Hunan Province of China (No. 2021JJ50107, 2022JJ40121), Education Department of Hunan Province of China (No.21B0657) and Hunan Students' Platform for innovation and entrepreneurship training program (2021 No.3297).

**Data Availability Statement:** Not applicable.

**Conflicts of Interest:** The authors declare no conflict of interest.

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
