# Peer review of "Achieving Tunable Microwave Absorbing Properties by Phase Control of NiCoMnSn Alloy Flakes"

_metals, doi:10.3390/met12091542_

Round 1

Reviewer 1 Report

The paper of Sun et al. describes in detail the microwave absorption performance of alloy flakes of the Ni43Co7Mn39Sn11 composition that, depending on the temperature treatment, show variable reflection loss reaching -56.4 dB at 8.8 GHz. The results are quite important and match the scope of the journal. This paper can be accepted after minor revision. I have only minor comments:

11. Something is wrong with the inner diameter on page 2, line 72. Please correct.

22. The authors state that the S-6h sample is a mixture of L21 and gamma phases as confirmed by Figure 2, but what are the clearly visible unindexed peaks at ca. 57 and 63 degrees 2-theta?

33. Also in Figure 2, the S-6h pattern, the 311 index is obviously incorrect. It should be 511 or 333.

44. What is “j” in formula 1?

55. The legend to Figure 6 is misleading. Whereas the RL versus f and d(calc) versus f are indeed curves, the d(m) versus f is not a curve but a set of points.

Author Response

Dear Reviewer,

Thanks for your valuable comment.

Best wishes

Author Response

(The authors gave the same response as above.)

Reviewer 3 Report

    The microwave absorbing properties of the composite containing Ni43Co7Mn39Sn11 alloy flakes in the paraffin matrix have been studied in frequency range from 2 to 18 GHz. Flakes were fabricated by balling milling method, and the contents of γ phase in the flakes were tuned by the subsequent heat treatment process. The XRD and SEM characterization of the flakes has been performed. After fabrication and thermal treatment the composite Ni43Co7Mn39Sn11 alloy flake-paraffin matrix has been obtained.  The composite samples exhibited excellent tunable microwave absorption. The optimal reflection loss was achieved to -56.4dB at 8.8 GHz with a single thickness of 2.0 mm. As the authors believed, good absorbing properties can be achieved due to the optimal structure of Ni43Co7Mn39Sn11 alloy flakes by phase control. The favorable combination of permittivity and permeability values secures better impedance matching. This study proved that Ni43Co7Mn39Sn11 alloy flakes should be a promising microwave absorption material. The paper, however, cannot be published in the presented view.

  1. In Fig.3a there is a peak value in ε’ at 15.8 GHz for the S-0h sample. What is the reason for the peak and why it is absent for the other samples?   Could you present at least a supposition why “the higher level of μ’ and μ” for the three samples are reversed to their values of ε’ and ε” “?.
  2. What is the reason for the real part of magnetic permeability to take on an unusual value less than 1 for sample S-0h?
  3. It is spoken at p.4 line 107: “sample S-6h exhibits the minimum value of ε’ and ε”.  However, there is no minimum for sample S-6h in Fig.3b.
  4. The authors several times wrote that Ni43Co7Mn39Sn11 alloy flakes exhibited excellent microwave absorption properties. In fact, the microwave absorption of the composite but not pure flakes is studied in the paper. This substitution is especially inappropriate in Conclusions. 
  5. In the text after formulas (3) and (4) the repeated definitions of the designations are given.
  6. The misprint at p.2 line 72 has to be corrected.

Author Response

(The authors gave the same response as above.)

Round 2

Reviewer 2 Report

The authors have answered all the questions raised and have made the changes accordingly. Congratulations to the authors for their work.

Author Response

(The authors gave the same response as above.)

Reviewer 3 Report

The revised manuscript is suitable for publication with minor corrections.  "Gilbert"  should be printed in line 91. In line 92 "the dynamic initial permeability" should be printed because this quantity is frequency dependent. 

Author Response

(The authors gave the same response as above.)
